# Hearing Care: Safe Listening Method and System for Personal Listening Devices

**DOI:** 10.3390/ijerph20032161

**Published:** 2023-01-25

**Authors:** Fei Chen, Hui Xue, Meng Wang, Zhiling Cai, Shipeng Zhu

**Affiliations:** 1School of Computer Science and Engineering, Southeast University, Nanjing 211189, China; 2MOE Key Laboratory of Computer Network and Information Integration, Southeast University, Nanjing 211189, China

**Keywords:** dose calculation, hearing loose, safe listening device, safe listening APP

## Abstract

Excessive use of Personal Listening Devices (PLDs) and prolonged exposure to noise from loud music create many potential risks associated with hearing loss. To this end, the World Health Organization has published Recommendation ITU-T H.870 in 2019, which provides adults and children with a set of recommendations for sound dosage and operating times needed to avoid potential hearing risks. Some studies have investigated noise exposure of related applications for listening safety, resulting in some related recommendations and applications; however, these studies often do not pay attention to measurement error, which is important for human real noise exposure estimation to avoid hearing loss. This paper proposes a method for calculating noise exposure that can accurately calculate the actual noise sound-pressure level (SPL) and PLD dosage based on the WHO-ITU standard. We develop a calculation method and design a listening system that includes (i) a Safe Listening Personal Listening Device (SL-PLD) that can measure the listening dose in real time and control the output volume effectively, (ii) a Safe Listening Application (SL-APP) for assisting the SL-PLD to check the listening status in real time and provide alerts. Our experimental results show that (i) the proposed noise calculation method can reach 0.88 dB deviation under the 76 dB reference SPL and 98.8% accuracy, as compared to the SoundCheck tool measurement, (ii) the proposed SL-PLD controls the SPL output effectively as the dose increases, and (iii) the SL-APP determines the dosage usage and will provide a warning when the dosage exceeds a preset value. Therefore, users can adjust their listening behavior for more secure listening by using our methods and applications.

## 1. Introduction

Personal listening devices (PLDs) are ubiquitous in daily life when listening to music, making phone calls, etc. Long-time exposure to such devices increases the risk of permanent hearing loss [1,2]. Many studies on PLDs usage have shown that nearly half of the surveyed teenagers and young adults are at risk of unsafe exposure to continuous loud volume during the usage of PLDs, while only a few of them have safe listening habits or attitudes [3,4,5]. Furthermore, the World Health Organization (WHO) has estimated that more than 1.5 billion people worldwide are at risk of some form of hearing loss due to unsafe listening practices [6].

Several studies have shown that long-term exposure to loud levels caused by music or noise can lead to auditory symptoms such as temporary hearing loss, listening shifts, and permanent hearing loss [7,8]. To effectively assess the impact of noise on people, some international organizations have promulgated a series of standards. Standards were created by US organizations such as the National Institute for Deafness and Communication Disorders (NIDCD), the National Institute for Occupational Safety and Health (NIOSH), and the Occupational Safety and Health Administration (OSHA) [9] to promote safe listening and reduce the risk of noise-induced hearing loss (NIHL). The NIOSH standard recommended exposure level is 85 dBA over 8 h daily. In contrast, the OSHA standard permitted noise exposure limit is 90 dBA over 8 h daily. WHO, in cooperation with the International Telecommunication Union (ITU), proposed a dose exposure standard WHO-ITU H.870 to protect youth and adults from over exposure to music noise caused by unsafe use of PLDs [10,11]. For occupational noise, we can only measure and protect hearing loss through methods such as wearing earmuffs, etc. However, when using PLDs to listen to music, we can design specific audio devices and apply algorithms to control the loudness of music automatically to achieve safe listening, even if the user does not have the sense of safe listening. 

Firstly, we define accuracy as the max absolute deviation of SPL read on SL-APP compared to the value measured by acoustic equipment. How to estimate sound dose accurately in a safe listening system, including the use of headphones and software APPs to keep listening safe or tracking exposure status accurately, is important. The sound dose is defined as the total quantity of sound received by the human during a specified period [11], which provides guidelines for safe listening with PLDs. When sound originates from a speaker close to the ear or inside the ear canal, then the sound exposure depends on the frequency response characteristics of the speaker from the input signal to the eardrum, which should be accounted for or corrected in methods that aim to calculate the noise exposure directly from the music sound trackers or digital input signals from the PLD. When using an APP to show dose exposure status, on the condition of ignoring geometric sound dispersion, it is misleading for users to track their noise exposure status because the real SPL near the ear drum is not comparable with the APP. Taking Figure 1 as an example, the sound signal from path B makes +3 dBA loss from 85 dBA and gets 88 dBA at TP (test point), which is fixed at the external ear, compared to path A, which reaches the ear directly. Long-time listening with such SPL deviation may cause dose overflow and increase listening disease risk. However, estimating dose accuracy as far as possible can alleviate this problem.

Based on the theory described above, this study mainly focuses on how to calculate sound dose and build a safe listening system. To achieve this target, we can summarize our novelty study in three aspects:We proposed a method to accurately calculate the sound dose for music noise and occupational noise.We designed a SL-PLD and a SL-APP and described how they work in detail. For the first time, we used a clock chip in the SL-PLD to align noise exposure time with RTC.We described the test methodology for evaluating the safe listening system, which includes the SPL accuracy measurement, the reliability of SPL control, and the display of daily and weekly dose information on the SL-APP.

In PLDs, the measurement noise exposure of music has a strong relationship with Speakers because of the discrepancy of the response curve, which makes it difficult to take accurate measurements. To alleviate this problem, we take the Frequency Response Characteristics of Speakers into consideration, getting more accurate noise exposure SPL and dose value, for the first time to explore the calculation method in the frequency domain in detail. Unlike music noise, to measure occupational noise exposure, we need a Microphone and consider its response-curve discrepancy to make the measurement more accurate. Based on the calculation theory, we designed a safe listening system, including a SL-PLD with dose calculation and control system and a SL-APP with SPL and dose notification, to protect users from hearing loss via music noise exposure control rather than influencing the listening experience. In detail, we consider the WHO-ITU H.870 safe listening standards and the sound reference exposure dose formula based on the time sound level. To calculate dosage and SPL accurately at ear drum around, A-weighted [12] and diffuse-field corrected sound pressure and speaker sound level curves should be taken into account. We use Dynamic Range Control (DRC) [13] arithmetic to control SPL flexibly when the exposure dose is above the reference value or SPL is too loud. The APP designed with real-time and accumulated exposure information expression connects with PLD and obtains historic exposure information from BT at first connection, helping create a user-centered safe listening experience of SPL measurement and dose calculation.

The remainder of the paper is organized as follows: Section 2 describes related works on safe listening. In Section 3, we describe the methods of dosage and SPL calculation and the design of a safe listening system that controls sound exposure on PLD. Section 4 describes the experimental validation of our work. We conclude our work in Section 5. We select representative music tones and compare the calculated results with the acoustical instrument and conclude that the calculated values are very close to the measurement results. By using the designed headsets over many weeks, with exposure levels that are less or more than the theoretical reference, we find that the SPL is effectively controlled.

## 2. Related Works

Under the guidance of different safe listening standards, many applications (APPs) have been developed for noise measurements or hearing loss detection. Hear-WHO, proposed by WHO, can help people who often listen to loud music, work in noisy places, etc., to facilitate hearing checks, track changes in hearing status, and help people notice any hearing loss as early as possible [14]. The NIOSH Sound Level Meter (SLM) app was developed to measure occupational noise, helping workers understand more about the noise environment and provide guidelines for hearing health [15]. Health App was developed to integrate U.S. and United Nations (UN) safe listening standards and to serve healthcare environments [16]. It is provided as a tool to measure users’ daily sound and ambient noise exposure through the personal audio system. Study [17] realized the Hearing Health app, which integrated functions for different noise exposure standards, such as daily noise exposure measurement, sound dose estimation, risk notifications, etc. Paping et al. developed a smart application running on Android devices to objectively measure music listening habits by calculating music dose exposure status among adolescents; the results revealed that about 20% of them are at risk of over 50% of WHO-ITU weekly dose standard [18].

Most APPs [15,16,17,18] focus on how to integrate different standards and hearing-loss checks [14], while few studies focus on noise exposure calculation accuracy and hearing protection systems. In addition, current APPs do not have effective methods to control sound exposure, especially when people listen to loud music with headsets at high sound volumes. As described in Section 1, in contrast with the methods in [15,16,17,18], our study first describes how to calculate noise exposure accurately, followed by discussing the SL-PLD and safe listening system design method. We describe our results in the following sections.

## 3. Methods

The WHO-ITU standard stipulates weekly dose exposure by listening devices for adults and children. It is recommended that, for safe listening, adult weekly dose exposure should be less than 1.6 Pa^2^h per 7 days and that of children should be less than 0.51 Pa^2^h. Noise exposure standards proposed by CDC, OSHA, US ARL, etc., recommend the safe noise exposure sound-pressure level (SPL with dBA) and time. We first proposed an accordingly accurate dosage/SPL calculation method that can be applied to PLD. Based on the calculation, our designed SL-PLD system can effectively measure and control sound exposure, preventing it from exceeding the WHO-ITU standard. Moreover, the method can also be used to measure noise exposure status when applying CDC, OSHA, US ARL, etc.

### 3.1. Dosage and SPL Calculation

A-weighted is the most common curve defined in the International standard IEC 61672:25. It is broadly applied to PLDs and occupational noise sound-level measurement to assess potential hearing damage. The A-weighted calculation [19] is described as
(1)$$A\left(f\right)=10lg{\left[\frac{f_{4}{}^{2}f^{4}}{\left(f^{2}+f_{1}{}^{2}\right){\left(f^{2}+f_{2}{}^{2}\right)}^{1/2}{\left(f^{2}+f_{3}{}^{2}\right)}^{1/2}\left(f^{2}+f_{4}{}^{2}\right)}\right]}^{2}dB-A_{1000},$$
where $f_{1}=20.60\;\mathrm{Hz}$; $\;f_{2}=107.7\;\mathrm{Hz}$;$\;f_{3}=373.9\;\mathrm{Hz}$; $f_{4}=12,194\;\mathrm{Hz}$; and $A_{1000}=-2.0\;\mathrm{dB}$.

Knowing the A-weighted values, we can calculate the dosage, which is the integral of sound exposure over a time [11]. Mathematically, it can be expressed as
(2)$$Dose=\int_{t_{1}}^{t_{2}}p_{A}{}^{2}\left(t\right)dt,$$
where $p_{A}{}^{2}\left(t\right)$, $t_{1}$ and $t_{2}$ are the square of the A-weighted sound pressure, $t_{1}$ the exposure of the start time and the exposure end time, respectively. For a PLD such as a headset, the analog signal is sampled to digital data via its frequency $f_{s}$, and is processed frame by frame. The frame size N can be adjusted based on requirements to 20 ms or 25 ms (data size). Therefore, Equation (2) can be rewritten as
(3)$$Dose=\sum_{i=0}^{N-1}{\left|x{\left[i\right]}_{A}\right|}^{2}*∆t$$

Since the A-weighted calculation A(f) is in frequency domain, it is difficult to calculate the sampled audio signal $x{\left[i\right]}_{A}$ in the time domain. Based on the well-known Parseval’s theorem [20], we can calculate the discrete A-weighted audio signal $x{\left[i\right]}_{A}$ in the frequency domain. We define A[k] as the A-weighted signal at frequency $f=k*f_{s}/N$, set $X\left[k\right]$ as the audio frequency domain signal, obtained by STFT; then, Equation (3) can be rewritten as
(4)$$Dose=\frac{1}{N}\sum_{k=0}^{N-1}{\left|X\left[k\right]*A\left[k\right]\right|}^{2}*∆t.$$

The dose calculation in Equation (4) cannot be applied to PLDs yet because the speaker frequency curve of different PLDs is not the same. The diffuse-field corrected sound pressure should also be taken into consideration, as it is proposed by WHO-ITU. Given a customized PLD, the speaker frequency power curve of instrument-measured sound levels can be expressed by the frequency response from frequencies 20 Hz to 20 kHz. Recommendation ITU-T P.58 states the tolerances on the head and torso simulator (HATS) diffuse-field frequency response, which includes the tolerances in the calibration of the occluded-ear simulator [21]. Additionally, it can be transformed into a signal table via an inverse operation. We define $S\left[k\right]$ as the speaker frequency signal and $D\left[k\right]$ as the diffuse-field frequency signal and align their size with $X\left[k\right]$. A more accurate dose calculation of this PLD is given by
(5)$$Dose=\frac{1}{N}\sum_{k=0}^{N-1}{\left|X\left[k\right]*A\left[k\right]*S\left[K\right]*D\left[K\right]\right|}^{2}*∆t.$$

Using vector expressions, Equation (5) can be simplified to:(6)$$Dose=\frac{∆t}{N}X_{S}{}^{T}X_{S}{}^{*},\;$$
where $X_{S}=X⊙A⊙S⊙D$ and *N* is the vector size. The symbol ⊙ means vector multiplication. Generally, the notation N is the SFTF frame size.

Equation (6) is an accurate dose calculation for PLDs such as Headsets or EarPods with a Microprogrammed Control Unit (MCU) controller. However, occupational noise exposure calculation is a little different, since such noise reaches the ear directly by air and not by speaker. To calculate occupational noise exposure, we can use a microphone on PLDs outside the ear to sample noise signal for dose calculation. Like the calculation of the music dose, it needs to consider the attenuation or enhancement at some frequency band when sampling the noise signal, as the compensation in the calculation.

The NIOSH recommended noise exposure dosage should not exceed 100, otherwise it is hazardous. This can be calculated by
(7)$$D_{NIOSH}=\left[\frac{C_{1}}{T_{1}}+\frac{C_{2}}{T_{2}}+\ldots +\frac{C_{n}}{T_{n}}\right]*100,$$
where $C_{n}$ represents the exposure time at the specified noise level. $T_{n}$ is the max time duration that should not be exceeded during the specified noise level.

The exposure can be measured by 85 dbA over 8 h noise exposure, which is calculated via the following formula
(8)$$T\left(min\right)=\frac{480}{2^{\left(L-85\right)/3}}\;,$$
where L is the exposure A-weighted level and 3 is the exchange rate. $T\left(min\right)$ is the reference of T_n in Formula (7).

To determine noise exposure status in accordance with occupational stands, e.g., NIOSH, OSHA, etc., the key point is to calculate the noise level L. To consider the microphone discrepancies between the different audio sampling devices, we measure the microphone sound levels from the speaker and convert these values to a signal vector defined as M and $X_{M}=X⊙A⊙M⊙D$. The dosage calculation is obtained via sampling from the microphone. This is expressed as
(9)$$Dose_{os}=\frac{∆t}{N}X_{M}{}^{T}X_{M}{}^{*}.$$

Mathematically, the noise level *L* is given by
(10)$$L=10\;\mathrm{lg}\left\{\frac{Dose}{T*p_{0}{}^{2}}\right\}dBA,$$
where *T* is the audio frame time and $p_{0}{}^{2}$ is the reference sound pressure of 20 μPa.

Standards from CDC (NIOSH) and US ARL (Military) prescribe continuous noise exposure over a prespecified amount of time. We suggest using portable devices with microphones (such as a smart watch, smart phone, or earbuds) for noise-exposure estimation.

### 3.2. Safe Listening System

Based on the arithmetic introduced in Section 2, we designed a safe listening system, which includes safe listening headphones SL-PLD and SL-APP. The SL-PLD can work in modes offline, e.g., lost connection with APP, and online, e.g., connected with APP. Such design mode has two advantages: (1) In offline mode, the SL-PLD automatically adjusts according to the usage of the dose to protect the user from dose exposure for too long. We stored dose and SPL information every second at the SL-PLD side for exposure status reference when connect with SL-APP, which is not available in other PLDs. (2) In online mode, the SL-PLD interacts with the SL-APP and uploads listening status information in real time so that users can easily understand their listening status. In the following content, we introduce our safe listening system from hardware and software in two aspects. The hardware refers to listening earphones with a Bluetooth chip with ARM Cortex-M4F as the main calculation core, and the software includes the safe listening software running in the SL-PLD and the SL-APP running on the mobile phone.

#### 3.2.1. SL-PLD Hardware Architecture

The hardware structure of the headset is not complicated, as shown in Figure 2. It includes many parts. We adopt a Bluetooth chip with an ARM Cortex-M4F core [22], whose frequencies in bursts are 208 MHz. We run dose-calculation software in the Cortex-CM4 processor and apply an audio encoder and decoder in DSP. We then use BT to transfer the music signal and communicate with SL-APP under SPP and BLE protocol. We save SPL, dose, time, etc., information every one second, which takes about 16 bytes. As a rough estimate, based on the header information of the saved data, the flash supports about a one-year dose, and SPL storage is long enough for history exposure information review. The audio sensors include a speaker and an analog microphone. The speaker has parameters 4 ohms and 4 watts, and the microphone sensor is highly sensitive. The codec is an audio signal transformer, which converts digital signals to analog signals for speaker and converts analog signals from microphones to digital signals for algorithmic processing. To align dose and SPL information with exposure moment, we innovatively use real-time clock (RTC) in SL-PLD. This is a very important peripheral because the calculated dose information needs to align with Greenwich Mean Time (GMT), such that the user can obtain information about when the exposure is high. To make the SL-PLD working time longer to improve the user experience, we use an 800 ma battery on our SL-PLD, allowing the device to run continuously for up to three days.

#### 3.2.2. SL-PLD Safe Listening Software System

The safe listening software system is a set of software that includes the front-end and the back-end, serving as listening protection and alert functions. As shown in Figure 3, the front-end software works on listening devices such as headsets for safe listening protection, and the back-end software is an App that runs on mobile phones with Android or iOS operating systems. The transplantable part in the front-end software only works for noise exposure calculation and can work on most portable headset devices with RTC and microphones. The App displays sound exposure information from the headset via BT. By default, we apply the WHO-ITU standard, e.g., 1.6 Pa^2^h weekly for adults and 0.51 Pa^2^h weekly for sensitive users such as children, in our safe listening software system. It requires a specific configuration to enable occupational noise measurement, since occupational noise measurement needs the microphone to work continuously for more than 8 h (suppose the working time is 8 h), which leads to a shorter SL-PLD working time for music playing. In practical applications, we almost design specific low-power hardware and software design for occupational noise measurement application scenarios. Next, we introduce our safe listening system for details.

There are multiple audio input channels on a headphone, including AUX music, Bluetooth music, Bluetooth phone, and microphone recording. AUX music, Bluetooth music, and Bluetooth phone are labeled as music signals, while signals recorded by the microphone are labeled as noise signals. The music signals from different channels are transmitted to the ear through a speaker, generating controllable sound dose exposure. Referencing the WHO-ITU standards, we split the Adult Dose (1.6 Pa^2^h) and Child Dose (0.51 Pa^2^h) in a listening period to many thresholds for SPL control. The acoustic signal is compressed to lower the SPL via the DRC algorithm when one threshold is reached and to control the sound dosage. However, the noise signal recorded by the microphone is uncontrollable and can only be measured and tracked. When this function is enabled, the noise dose for this environment is calculated and recorded on PLD. All healthy listening and control algorithms on the earphone running on DSP calculate music exposure and occupational noise in parallel. When dose exposure calculation is working, the measurements are saved every second to align with the GMT provided by RTC.

The APP is designed as a safe listening assistant that prompts users to observe the music listening status or noise status acquired from the headset via the Control Protocol Message (CPM) when it is opened. Meanwhile, it can make decisions such as lowering the SPL. This is especially useful if parents want to restrict the SPL to a lower threshold when children use the SL-PLD. The APP also provides safe listening recommendations and descriptions to help users use SL-PLD safely, as well as providing alerts with different colors when noise exposure reaches different levels.

The APP tool is mainly composed of four modules, including daily-dose usage, weekly dose usage, noise exposure, and alert displays. The CPM-wrapped exposure information follows the Bluetooth BLE/SPP protocol between the app and headset. When the WHO-ITU standard is selected, module A displays real-time SPL information and module B displays weekly dose exposure information. When switching to the occupational noise measurement standard (CDC, NIOSH, OSHA), the occupational noise module displays the noise exposure status throughout the whole day. The alert module uses the colors sky blue, yellow, and red to alert the exposure status to users.

## 4. Experiments and Results

SoundCheck is a powerful and flexible audio test system which is used to measure different kinds of audio devices, from simple transducers to complex Bluetooth, etc. Its advanced measurement algorithms ensure ultimate accuracy, and the measurement sequence is simple and fast [23]. To validate the proposed dose calculation method and the listening protection function of the safe listening system, we conducted a simulation to test various factors: (1) getting an A-weighted Value and Frequency Response Curve Bias method for Dose calculation, and using the SoundCheck software platform to measure SPL output; (2) verifying the function of volume compression when dose exposure grows to achieve listening protection; (3) determining dose and SPL information via the SL-APP for intuitive understanding of noise exposure.

Environment setup. We connected an artificial ear to the SoundCheck System via a signal converter. To test the SPL accuracy, an SL-PLD was attached to an artificial ear connected to a SoundCheck acoustic measurement tool. The SL-PLD connected with the SL-APP via BT. Each tone was played at different volumes and the real output in SoundCheck was measured to verify the SPL accuracy. We used the left artificial ear to test one speaker of the SL-PLD and then switched the other speaker to this artificial ear and performed the test again. In our test, both speakers measured at the left artificial had almost the same SPL, and the SPL at each volume demonstrated almost no change. We tested 10 times and averaged the SPL value to determine the test result. For DRC and exposure status testing, we connected the SL-PLD and play tones continually and observed the outputs. The following sections describe our experiment setting in detail.

### 4.1. Diffuse Field and A-Weighted Value

Discrete field frequency response is commonly used for audio signal calculation. The International Telecommunication Union (ITU) ITU-T P.58 provides a head and torso simulator (HATS) occluded-ear simulator with 24 frequency points of a discrete field frequency response [19]. Considering the hardware limitation and calculation accuracy during the actual calculation, we used a 512-point STFT calculation, so we used the interpolation method to expand the frequency response of the scattered field to align the length with STFT size. For the A-weighted frequency response [19], we used Formula (2) to generate 512 weights directly.

### 4.2. Get Speaker Frequency Response Curve Bias

To get the SPL and dose as exposed near the human ear, one of the keys is to get the speaker frequency response bias. We configured the headphones to only run with an audio equalizer filter algorithm (EQ) and to connect to the acoustic measuring instrument SoundCheck, which runs a 0 dB sweep signal with a frequency range from 20 Hz to 20 kHz. The frequency response diagram of the headphone speaker is shown in Figure 4. When conducted this experiment, we used the left artificial ear to test one speaker, and after that we switched another speaker to the left artificial ear for testing. We found both speakers have almost the same frequency response, so we just adopted the left speaker’s frequency response for calculation. We ran this operation five times with the same headphone to obtain the average value. The final speaker frequency response curve calculates the bias needed to correct the SPL.

### 4.3. Dose and Loudness Testing

As shown in Equation (2), the dose can be calculated from the sound-pressure level without any loss. We only measured the real output SPL and confirmed it was comparable with the target preset expected SPL in this experiment. If the calculated SPL was comparable with the one detected by acoustic instrument, we could conclude that our dose calculation result was accurate.

As our SL-PLD performance is not good when the frequency is over 8 kHz because of the hardware Speaker attribute (as shown in Figure 3), we did not consider signals with a frequency over 8000 Hz in this experiment. Hence, we connected headphones with PC by BT and played a tone with frequency groups (63 Hz, 100 Hz, 125 Hz, 200 Hz, 250 Hz, 300 Hz, 400 Hz, 500 Hz, 1000 Hz, 2000 Hz, 4000 Hz, 8000 Hz), playing a 0 dBFS signal. We set the expected target output loudness for the SPL group (75 dBA, 76 dBA, 78 dBA, 80 dBA, 82 dBA, 84 dBA, 86 dBA). We measured and recorded the SPL via SoundCheck with the hardware environment setting, as shown in Figure 5a. We placed the SL-PLD on an artificial ear that connects with acoustic equipment to receive audio signals via the PC. We then played the preset audio signal above and read the real output SPL on the SoundCheck SPL measurement window. In Figure 5a, we used a different physically structured SL-PLD with the same speaker compared with the one in Figure 2, and determined that the physical structure has no effect on the result.

The measured results are shown in Figure 5b. With frequency arranged from 60 Hz to 8000 Hz at the selected frequency point, the measured SPL is very close to the target SPL (dotted curve), and the max mean bias is about 98.8% (measured error 0.88 dBA at reference SPL 76 dBA), which means our proposed method can reflect the real dose exposure of the human ear explored as much as possible. We tried to to measure the output SPL with the signal at 16,000 Hz but our result was at least about 6 dBA lower than the target SPL. The details of our measured data are summarized in Table 1; it is obvious that SPL drops greatly at 16,000 Hz at each SPL level. Take target signal SPL at 75 dBA, for example; we just measured 67.8 dBA at the artificial ear measurement point. This phenomenon will alleviate when the speaker frequency response characteristic is better. In addition, the proportion of the frequency higher than 8000 Hz in most music is much smaller than the proportion that is lower, so we ignore the bias above 8000 Hz in our SL-PLD.

### 4.4. Safe Listening Control

To automatically limit the output volume base on dose exposure status, we used DRC to compress the volume in our Safe Listening System. Music SPL is compressed by the DRC algorithm when it exceeds a predefined threshold. To make the SPL control more intuitive, we used a children’s weekly dose of 0.51 Pa^2^h/10 for one level to adjust volume and obtain a dose threshold list (0.0051 Pa^2^h, 0.0102 Pa^2^h, 0.0153 Pa^2^h, 0.0204 Pa^2^h, 0.0255 Pa^2^h, 0.0306 Pa^2^h, 0.0357 Pa^2^h, 0.0408 Pa^2^h, 0.0459 Pa^2^h, 0.051 Pa^2^h), accordingly setting the volume change from 87 dBA to 69 dBA and getting a target volume list (87 dBA, 85 dBA, 83 dBA, 80 dBA, 78 dBA, 76 dBA, 74 dBA, 71 dBA, 69 dBA, 69 dBA). We played tone music with a continuous max volume over a set period of time and observed the changes in the SPL. The results are shown in Figure 6. Every time the dose exceeded the predefined threshold, the max SPL of music was effectively controlled by the DRC and suppressed to a predefined lower SPL paired with the dose. For example, when the accumulated dose reached 0.0153 Pa^2^h (index 3 in the dose list), the target volume was set to 83 dBA. Following this rule, we split the total dose more precisely for each day using 10 thresholds applied to SL-PLD.

### 4.5. Exposure Status Tracking

The SL-APP on the mobile phone mainly consists of two parts: the daily dose exposure and the weekly dose statistics. When powered on the SL-PLD, it connects to the SL-APP via Bluetooth protocol and uploads exposure information via the safe listening data transition protocol we defined. Then, the real time, the accumulated exposure, and the SPL information will display on the SL-APP. As depicted in Figure 7, the daily information presents SPL (dBA), lightning time, and uses percentage information that obeys WHO-ITU standard exposure allowances. The exposure information is updated every second. We split weekly exposure allowance between adults at 1.6 Pa^2^h and children at 0.51 Pa^2^h into seven levels and split daily dose exposure to six linear levels. When dose exceeds the predefined threshold of each day or one week, a warning labeled in yellow alerts the user to be aware of their listening status. When exposure reaches 100% each day or week, a red signal alerts the user to danger. The daily and weekly exposure, described in Figure 7 and Figure 8, respectively, provides the listening time, the dose, and the listening period exposure percentage for SPL compression. The figures use yellow and red colors to alert when an exposure threshold is reached, and the SPL is accordingly compressed to a lower level. As the read figure in Figure 7, the target SPL is compressed to about 75 dBA after exposure over the max permitted dose. In addition, when the dose exceeds each preset threshold, a warning message will pop up on the user’s mobile phone to remind them to adjust their listening habits. It is obvious that through cooperation with the APP and the earphone, the dose exposure of listening to music can be controlled intuitively and effectively to achieve the purpose of more accurate hearing protection.

## 5. Conclusions and Future

In this paper, we have demonstrated the effectiveness of calculating sound dose exposure in the frequency domain, and we have designed a SL-PLD and a SL-APP that protect users from unhealthy music-noise levels. The experiments proved the calculation method we proposed can determine an accurate SPL and dose, which can help control music noise exposure effectively. The SL-APP has intuitive components and an alert mechanism that can easily connect with the SL-PLD, which helps users determine their dose exposure status at any time and adjust their listening behavior when needed. However, we could only determine dose exposures on one device. When users listen with multiple devices in a listening period, the total dose exposure cannot be accurately determined. Additionally, as we do not have professional microphone testing equipment, we cannot currently provide an exact occupational noise experimental result. We will add the occupational noise measurement function in our safe listening system to make it more complete. In future work, we will (i) investigate how to use the user ID in cloud services to determine the listening dose when people use multiple devices, and (ii) determine how to present accurate, meaningful occupational noise experimental data.

## Figures and Tables

**Figure 1 ijerph-20-02161-f001:**
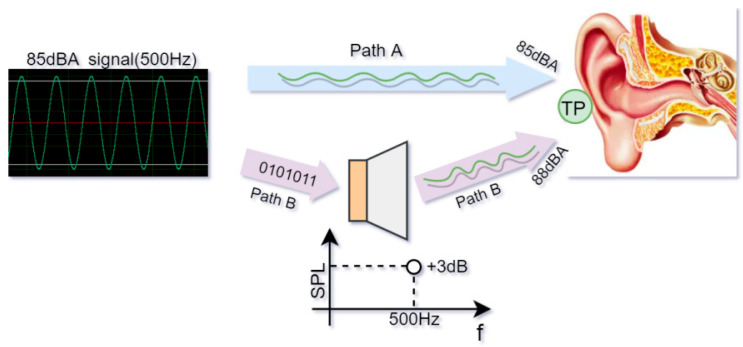
Sound-level deviation caused by the speaker makes dose estimation inaccurate. The speaker’s frequency response curve has a 3 dB deviation at frequency point 500 Hz.

**Figure 2 ijerph-20-02161-f002:**
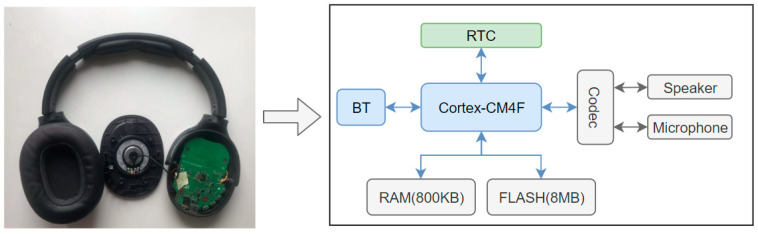
SL-PLD and Core-Board hardware architecture.

**Figure 3 ijerph-20-02161-f003:**
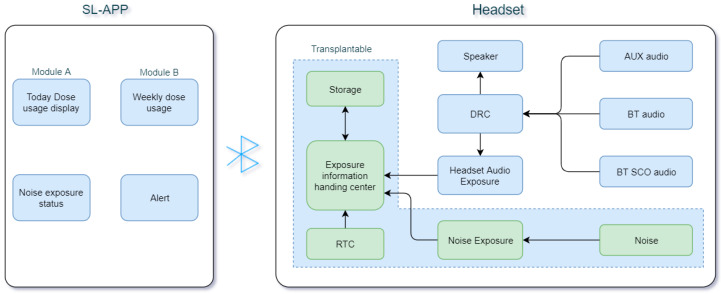
Safe listening software architecture in PLD.

**Figure 4 ijerph-20-02161-f004:**
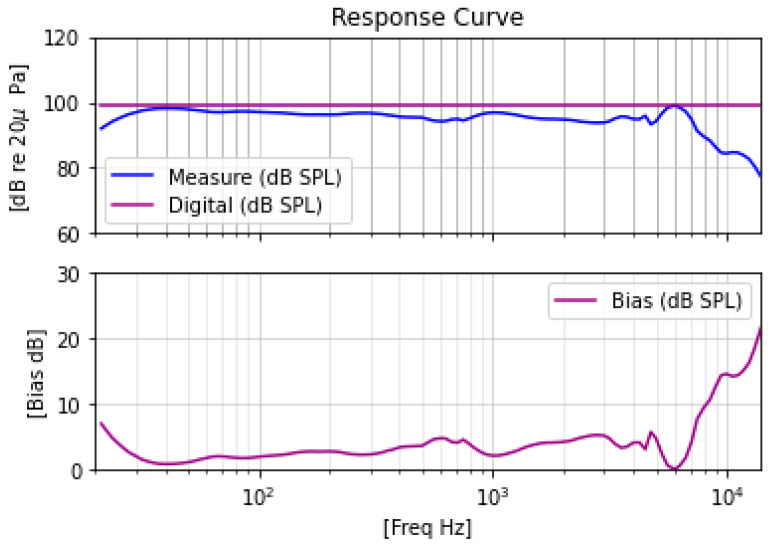
Speaker Frequency Response measurement and SPL bias for dose calculation.

**Figure 5 ijerph-20-02161-f005:**
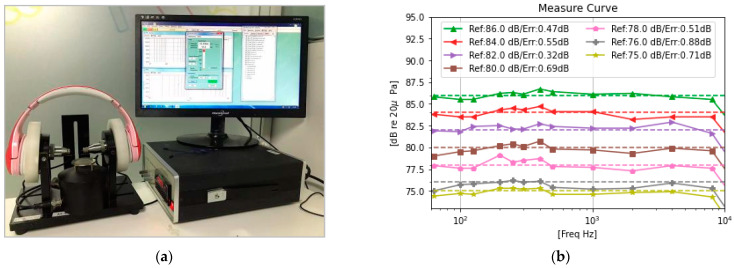
SPL Accuracy experiment platform and measurement result. (**a**) is the SPL measurement platform. Wear SL-PLD on artificial ear, connect with BT and play tone at different frequency and SPL; read the measured noise power. (**b**) is the real measured SPLs (dBA) from the measurement tool.

**Figure 6 ijerph-20-02161-f006:**
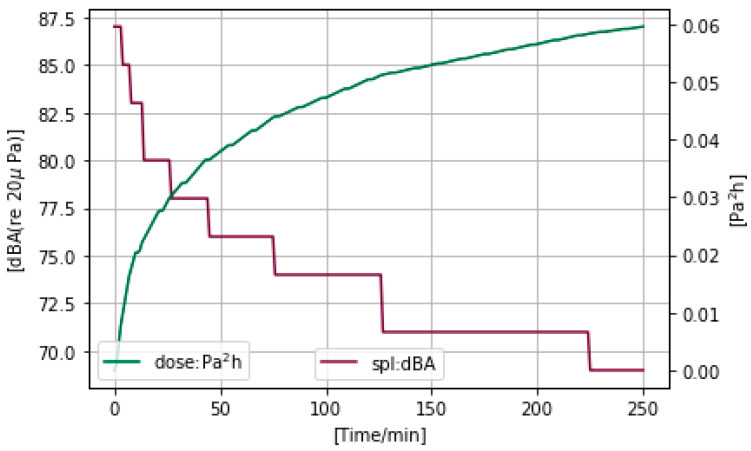
SPL suppression with growing Dose. With the growth of the accumulated Dose, the time spent at the target volume grows longer, which is good for the user’s listening experience.

**Figure 7 ijerph-20-02161-f007:**
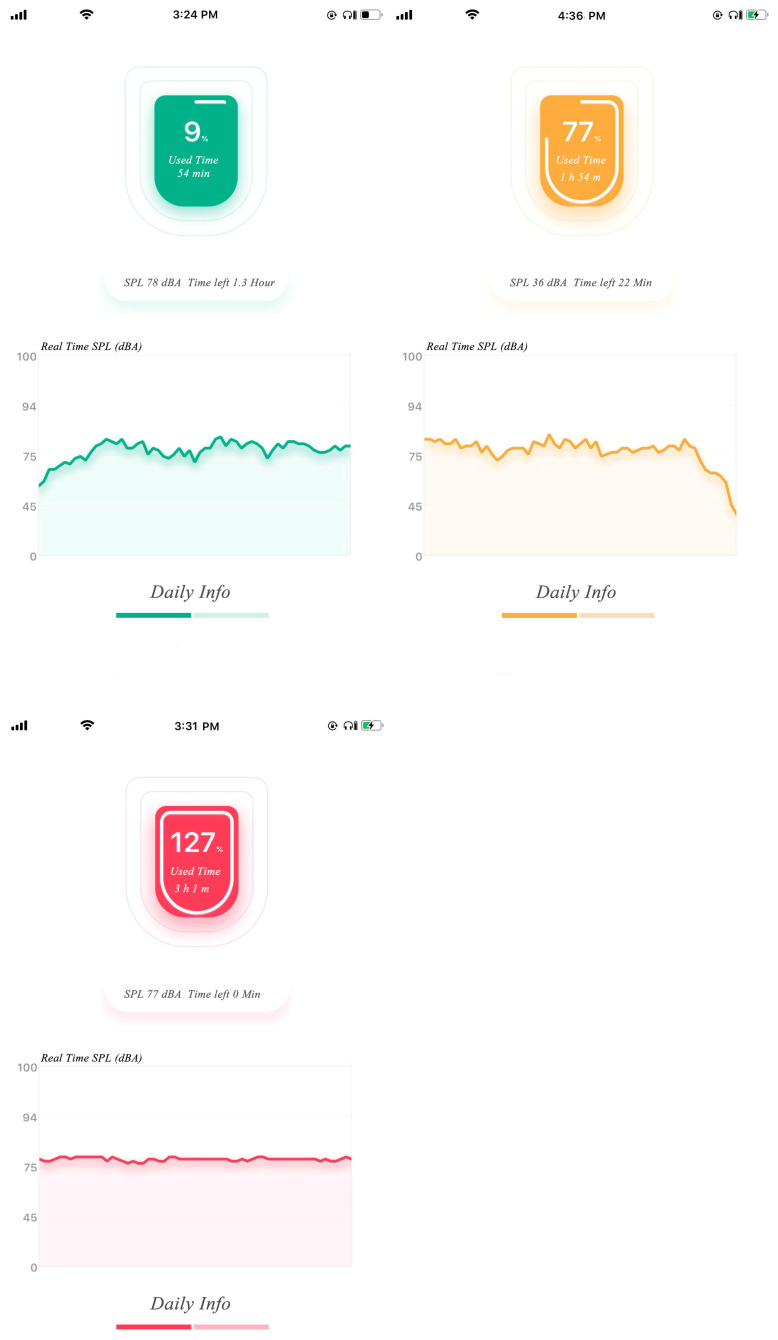
Daily dose usage status.

**Figure 8 ijerph-20-02161-f008:**
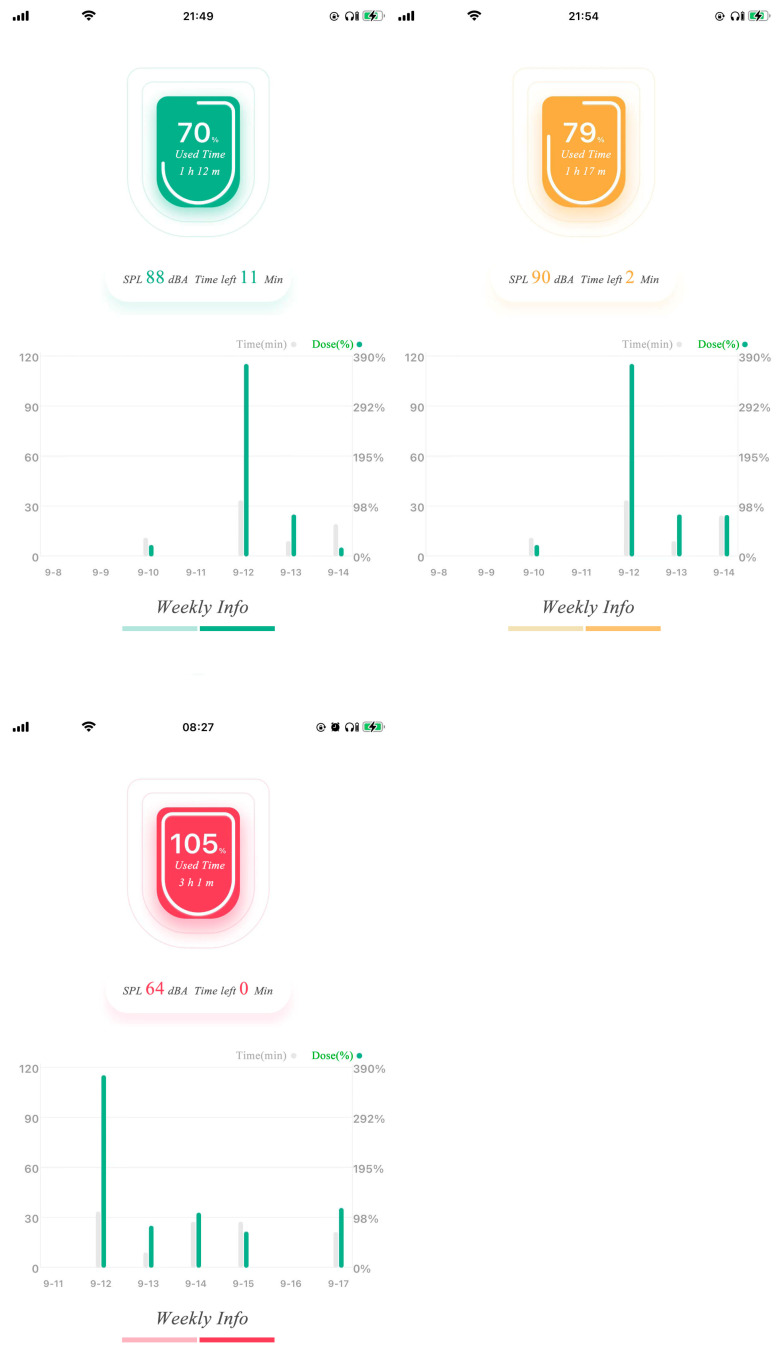
Weekly dose usage status.

**Table 1 ijerph-20-02161-t001:** The detailed data of SPL accuracy experiment. The first column is the reference SPL, and the data in the middle area show our measured SPL at artificial ear left side. Since our test was conducted in an open environment, not acoustic laboratory, there is measurement error at a single frequency point, but the mean is close to the reference SPL.

SPL (dBA) Ref\Test	Frequency (Hz)													Mean (dBA)
	63	100	125	200	250	300	400	500	1000	2000	4000	8000	16,000	
86	85.8	85.5	85.5	86.2	86.3	86.1	86.7	86.4	86.1	86.2	85.8	85.5	79.8	85.53
84	83.8	83.5	83.5	84.3	84.5	84.3	84.7	84.1	84.1	83.2	83.5	83.5	77.9	83.45
82	81.9	81.8	82.4	82.5	82.1	82.1	82.7	82.4	82.2	82.2	82.9	81.6	75	81.67
80	79	79.5	79.6	80.2	80.4	80.1	80.7	79.8	79.7	79.3	79.9	79.6	73.2	79.30
78	77.9	77.6	77.6	79.1	78.3	78.5	78.7	77.8	77.7	77.3	77.9	77.6	71.4	77.49
76	75	75.7	75.8	76	76.2	76	76.1	75.4	75.2	75.3	75.9	75.3	68.7	75.12
75	74.4	74.7	74.6	75.3	75.3	75.2	75.3	74.6	74.6	74.8	74.9	74.3	67.8	74.29

## Data Availability

The data are not publicly available due to privacy protection for essential workers.

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
