# Peer review of "Hearing Care: Safe Listening Method and System for Personal Listening Devices"

_ijerph, 2023, doi:10.3390/ijerph20032161_

Round 1
Reviewer 1 Report
The authors are addressing an important and difficult problem - how to measure noise exposure from Personal Listening Devices (PLDs) – and they describe a potentially-interesting calculation method. However, because of English language issues and a general lack of clarity on the methodological and technical aspects of the manuscript, it is difficult to assess the scientific value of this study. It is the opinion of this reviewer that many corrections are needed and possibly additional validation experiments.
The second paragraph in the introduction is confusing and needs attention. It starts by drawing attention to daily sounds like music, then goes on to cite occupational noise standards. These standards do not address the problem of PLDs as used in daily life. Also, note that the term “environmental noise” usually refers to non-occupational sources (e.g., nuisance from road traffic, recreational activities). Standard documents or recommendations from NIOSH and OSHA are for “occupational noise”, and it is a bit odd to say they help “to prevent mainly environmental noise induced hearing loss” (L42). I also do not grasp the intended message in the last sentence. Hearing protectors are primarily for protection against occupational noise, not environmental noise pe se, and in any case some advanced electronic hearing protectors do have features like compression and sound limiters to control sound exposure.
Figure 1 and accompanying text (L52-64) are also confusing and perhaps unnecessary. The +3 dB bias does not only depend on the “physical frequency response properties” of the speaker, but it also depends on the ear canal response from the speaker to the test point. The test point is undefined in the description. What do you mean by dose overflow and increase disease risk? The SPL bias from the speaker does not necessary lead to more exposure because users may compensate this effect by decreasing the device volume. All in all, it seems than the text and figure are a convoluted way of simply saying that when sound originates from a speaker close to the ear or inside the ear canal, then the sound exposure depends on the frequency response characteristics of the speaker from the input signal to the eardrum (or test point), and that this response must be accounted for or corrected in methods that aim to calculate the noise exposure directly from the music sound tracks or digital input signals from the PLD.
I find that Section 2 and Table 1 are incomplete and somewhat misleading. What are the methods/tools described in Refs 14 and 16 doing exactly? The methods in Table 1 seem to have different purposes and they output different things; how can they be compared one to one as in Table 1? Ref 13 is a method to monitor hearing using a digit speech test. It is not a noise exposure method (Line 108), so what does the check mark for noise exposure mean in Table 1. Ref 23 is said to be about listening habits (L113), again not noise exposure. If these methods have different purposes and outputs, how do you determine that they don’t have “accuracy” in Table 1 while your method is deemed accurate? You do not provide any definition for accuracy. I would consider Ref 13 to be quite accurate for what it does, and a lot of research has been published on such digit tests. Of note, the method by Nasrallah et al. 2019 (J. Acoust. Soc. Am. 145 (2), 749-760) seems related to yours (but different conceptually and methodologically) in that it attempts to calculate noise exposure from an internal speaker plus outside noise based on expected user behavior. As well, it may be worth considering in-ear noise dosimetry methods developed for hearing protectors e.g., Bonnet et al. 2020 (Applied Acoustics 157 Article 107015) or Nélisse et al. 2018 (https://www.euronoise2018.eu/docs/papers/101_Euronoise2018.pdf). The rating criteria for “Safe listening device” and “Safe listening system” are also not clear in Table 1.
The second half of section 3.1 on page 5 is not clear. It seems that you are proposing two methods and if so, please be more explicit. As I see it, the data from the speaker frequency response is used to calculate the music or signal (equation 5) exposure originating directly from the speaker of the PLD. Another method is sensing the noise inside the ear using a microphone and this method (equation 9) can be used to calculate the total exposure (= exposure from the PLD speaker plus the outside environmental noise that is transmitted into the ear canal). Instead, you refer to that second (microphone) method as calculating the ambient noise signal or the environmental exposure, which gives the impression that it calculates only the exposure arising from the noise around the user. Also, note that you do not say where this microphone is located, and it would be important to specify that it is inside the ear.
The experimental set up is not completely described. In Section 4.2, it would be important to specify which artificial ear is used to measure the speaker frequency response. Also, specify whether the five repetitions are for 5 different fit-refit or for only one headphone fit on the test set up? In Section 4.3, it is not clear how the target preset expected SPLs are obtained. Also, the artificial is again not specified so we don’t know if it is compatible with the one used in ITU-T P.58. The validation experiment in Section 4.3 (Figure 5/Table 2) seems to use the same stimuli (pure tones) and experimental set up as that used to obtain the speaker frequency response in Section 4.2. How can this then be an accuracy experiment (Table 2)? It seems you are simply cross-checking the data from Section 4.2. It seems that, for a proper validation, music soundtracks from different music genres would be used as stimuli to mimic daily life listening with the purpose of testing the accuracy of the noise exposure calculation method against the noise exposure from SoundCheck. Your method includes a diffuse field correction (Section 4.1), but in comparing your results to SoundCheck did you also apply the same diffuse field correction? Table 2 says that your method was compared to “measured SPL at artificial ear”, implying that you did not apply the correction on the measurement side.
I do not see data for the microphone response (term M on line 189) or any specific validation of the second method (equation 9).
Other comments or suggested edits:
L10-11: Poorly worded and unclear. Please rephrase.
L13: What do mean exactly by “statistical accuracy”? It may be better to speak of “measurement error” or “measurement uncertainty”.
L21: “99% accuracy” is unclear terminology when dealing with decibels because decibels are already a relative measure. The manuscript says that the max prediction error is 0.88 dB. It would be more instructive to report that dB max error value.
L28: I suggest “… are ubiquitous in daily life such as when listening to music, making phone calls, etc.”
L30: I suggest replacing “Many medical researches of” by “Many studies on”
L39: Please rephrase “credible standards” or delete “credible”. All standards are credible is some ways.
L45: I think it is 8 hours, not 9
L46: Spell “stand” in long.
L48-50: Unclear what the message is here. Please rephrase.
L54: I think Ref 11 is better described as providing guidelines for safe listening with PLDs, rather than being an exposure standard.
L57: I suggest “on the condition of ignoring geometric sound dispersion because …”
L59: sound “power”? I think it is sound “level” here if compared to 85 dBA.
L61: “loss”? It says +3 dB in Figure 1. Is it not a gain?
L66-67 Figure 1: This figure is not cited from the text. Sound power (level)? Overflow (deviation)?
L79: “Microphone”? Unclear since you only spoke about speakers at this stage in the paper.
L85: I suggest deleting “effectively”
L88-L89: More accurate than what? A-weighting and diffuse/free field corrections are absolutely needed or required if one wants to compare the calculated sound exposure with dose limits such as those specified in WHO-ITU H.870. It has little to do with accuracy or being understandable, it is a must.
L106: I suggest noise “measurements”
L111: “detection”? I suggest deleting the word here.
L112: I suggest “Paping [23] developed …”
L117: Check grammar and spelling (integrate, standard, loss)
L124: Table 1: See comments above.
L155: I think it is better to delete “instantaneous”.
L160-163: Not so clear. “Energy table”?
L170-171: PAD and MCU acronyms are undefined
L174: “microphone to simulate the ear”???
L177: You are using two definitions for “dose”. The one here from NIOSH and is in % and is generally the accepted definition. In the rest of the paper, dose refers to accumulated acoustic energy in Pa2-h.
L187-189: Not clear. Where is the microphone? In-the-canal? What does it mean “microphone sound levels from the speaker”? If the microphone is not near the eardrum, is it valid to use the diffuse field correction from ITU-T P.58?
L198: Again, are you describing devices with a microphone inside the canal?
L218: “transplant”? Do you mean “transfer”?
L244: Be more explicit. What did you use from the WHO-ITU standard? The maximum exposure doses in Pa2-h?
L245: Is it really “ambient noise detection” (present or absent) or is it “ambient noise measurement” (i.e., a dB value)”?
L261-265: Unclear how the two methods of calculating exposure work together. Presumably, the speaker method (equation 5) provides an estimate of the music exposure while the microphone method provides an estimate of the total exposure (not just the environmental noise per se)
L278-279: Again here, what do you mean by “environmental noise module”? It is unclear especially since NIOSH/OSHA are referred to in the literature as occupational noise standards, not environmental noise standards. It seems that the environmental noise module is a module that computes the TOTAL noise exposure which is a combination of the music signal exposure plus the outside environmental noise that gets into the ear canal.
L297: Diffuse field correction (not diffusion field)
L300: Please specify the exact ear simulator (presumably IEC 60318-4).
L309: More details are needed about the SoundCheck system used. Most importantly, where is the test point and which ear simulator is used? Is the experimental set up compatible to ITU-T P.58? If not, there is a mismatch, and it may not be adequate to use the diffuse field corrections from ITU-T P.58 (Section 4.1)
L311: Did you do 5 repetitions with the same headphone fit or did you refit the headphone each time?
L315: Figure 4 is not cited from the text.
L317-320: Unclear how targets are obtained.
L329: Which artificial ear? How many repetitions?
L331-332: Unclear. What do “outlook SL-PLD” and “outlook structure” mean?
L354: I stopped here
Author Response
Dear Editor,
We would like to thank the editors for giving us a chance to resubmit the paper, and also thank the reviewers for giving us constructive suggestions which would help us both in grammar and in depth to improve the quality of the paper. Here we explain the novelty and the significance of the paper in detail and submit a new version of our manuscript, which has been modified according to the reviewers' suggestions. Efforts are also made to correct the mistakes and improve the English of the manuscript. We mark all the changes in red in revised manuscript. We are looking forward to that these efforts would make our paper more acceptable for publication.
We appreciate your careful reading of our manuscript and valuable suggestions of the reviewers. We add coverletter first and follow the revised manuscript. Please review.
Besides, this manuscript is revised by English editor, sciworks, with two round, I can provide the revision history if necessary.

Reviewer 2 Report
There is much to like about this manuscript, as it addresses a public health issue which is both timely and highly relevant. Moreover, the authors have clearly dedicated significant time and effort towards developing approaches which could be used to address these public health issues. However, there appear to be a number of limitations which hinder my enthusiasm for this manuscript as currently written. One primary concern is that this approach seems to have been built with the idea that an individual is using a Bluetooth headset with an embedded microphone. This is a significant assumption and limitation, as it appears to have very little to offer if someone is not wearing such a device, and this possibility appears to have not been considered in the manuscript. A second major concern is that little consideration has been given to different types of headphones (e.g., earbuds vs circumaural headphones). Because of the proximity of the speaker to the eardrum, or the quality of fit to the ear, the SPL actually reaching the eardrum can vary despite similar input levels. As written, it doesn’t appear that this issue has been given any thought, which is problematic. Moreover, this issue may well influence the accuracy values provided in Table 2. Finally, I think the manuscript would benefit from more careful proofreading, as some of the sentence structure and wording was confusing and made the manuscript difficult to follow at times.
Specific comments are provided below.
Lines 36-50 – worth noting that those guidelines are not for PLDs, but for workplace environments.
Lines 45-47 - and what are those guidelines?
Lines 48-50 – run on sentence – general issues with sentence construction in manuscript
Lines 52-62 – this is confusing. All acoustic signals come from an origin source. Why does converting from A/D and passing through a speaker make things more intense unless that’s driven by the qualities of the speaker? If the latter, then there’s a simpler way to state that concept.
Lines 68-77 – there are a number of applications on the market. What is distinct about their approach? Is it more accurate?
Lines 78-82 – I agree in principle, but individuals can listen through a wide variety of devices and/or speakers. Thus, in theory one would need a nearly unlimited number of frequency response characteristics to maintain your desired degree of accuracy.
Lines 90-94 – wouldn’t this be driven heavily by the headpones / earbuds used with the PLD? It’s not clear how you’re planning on accounting for that.
Table 1 – you haven’t demonstrated accuracy yet, nor have you articulated why the other approaches may be inaccurate. This reads more like marketing than data.
Line 170-178 – from where are the microphone estimates taken? Near the ear? From the PLD?
Lines 237-243 – and if someone doesn’t use these Bluetooth headsets, does the system simply not work? How is ambient noise or sound level calculated then?
Table 2 – where are the measurements from the artificial ear made? The ear canal has a resonance characteristic which can influence the output. Moreover, depending on whether an earbud vs circumaural headset is used, it’s possible that different outputs could be observed. Wouldn’t your system potentially be more accurate if it could predict from an given input what the level of the ear would be given a type of earphone / earbud used?
Author Response

(The authors gave the same response as above.)
